# Whole genome sequencing of extended-spectrum β-lactamase genes in *Enterobacteriaceae* isolates from Nigeria

Christiana Jesumirhewe[1]*, Burkhard Springer[2], Franz Allerberger[2], Werner Ruppitsch[2]

**1** Department of Pharmaceutical Microbiology, Prof Dora Akunyili College of Pharmacy, Igbinedion University, Okada, Nigeria, **2** Institute of Medical Microbiology and Hygiene, Austrian Agency for Health and Food Safety (AGES), Vienna, Austria

☯ These authors contributed equally to this work.
* ebarunosen2002@yahoo.co.uk

**Data Availability Statement:** All relevant data are within the paper and its Supporting Information files. This Whole Genome Shotgun project has been deposited at DDBJ/EMBL/GenBank under the

## Abstract

Extended Spectrum β-lactamase (ESBL)-producing *Enterobacteriaceae* are of major concern as they are implicated in multidrug resistant nosocomial infections. They are listed on a recently published global priority list of antibiotic-resistant bacteria by the World Health Organization which raises concern in both healthcare and community settings. This study aimed at determining the frequency of ESBL genes in multidrug resistant human clinical *Enterobacteriaceae* isolates from Edo state Nigeria and to characterize the resistance mechanisms using whole genome sequencing. A total of 217 consecutive clinical isolates of *Enterobacteriaceae*, selection based on inclusion criteria, were collected from March-May 2015 from three medical microbiology laboratories of hospitals in Edo state Nigeria. All isolates were analyzed using matrix-assisted laser desorption ionization-time of flight (MALDI-TOF) mass spectrometry. Antibiotic susceptibility testing was performed by Kirby-Bauer method and minimum inhibitory concentration (MIC) determination by E-test method. Double disc synergy test was used to screen for the production of ESBL. Whole genome sequencing (WGS) was performed for isolate characterization and identification of resistance determinants. Out of 217 consecutive clinical *Enterobacteriaceae* isolates, 148 (68.2%) were multi-drug resistant. Of these multi-drug resistant isolates, 60 (40.5%) were positive for the ESBL phenotypic test and carried ESBL genes. CTX-M-15 was the predominant ESBL found, among 93.3% (n = 56/60). Thirty-two plasmid incompatibility groups and 28 known and two new sequence types were identified among the ESBL isolates. The high occurrence of CTX-M-15 with associated resistant determinants in multidrug resistant *Enterobacteriaceae* harboring different plasmid incompatibility groups and sequence types calls for the need of continuous monitoring of this resistance threat to reduce its public health impact. To our knowledge, this study presents the first genomic characterization of ESBL production mediated by $bla_{CTX-M-15}$ in human clinical isolates of *Enterobacter hormaechei*, *Citrobacter werkmanii* and *Atlantibacter hermannii* from Nigeria.

accession numbers JAAARR000000000-
JAAATV000000000.

**Funding:** This work was supported by an Ernst-
Mach grant 2015/2016 to C.J. by the Austrian
Federal Ministry of Science, Research and
Economy (BMWFW).).This work was supported
financially by the Austrian Agency for Health and
Food Safety (AGES), Vienna, Austria. The funders
had no role in study design, data collection and
analysis, decision to publish, or preparation of the
manuscript.

**Competing interests:** The authors have declared
that no competing interests exist.

## Introduction

*Enterobacteriaceae* can cause various infections and can also be found as commensals in the gastrointestinal tract. Beta lactam antibiotics are widely used for treatment of infections caused by *Enterobacteriaceae*. However, increased use of these antibiotics, particularly of third generation cephalosporins, is associated with the emergence of bacterial resistance caused by extended-spectrum beta-lactamases. Extended spectrum β-lactamases are enzymes capable of hydrolyzing a wide range of extended-spectrum β-lactams, including oxyiminocephalosporins and aztreonam, but are less active against cephamycins and carbapenems [1]. ESBLs are inhibited *in-vitro* by β-lactamase inhibitors such as clavulanic acid and tazobactam. Gram-negative bacteria producing ESBLs often acquire associated resistance to fluoroquinolones, aminoglycosides, tetracycline and chloramphenicol [2]. ESBL genes and quinolone resistance (PMQR) genes can be co-located either on the same plasmid or on different plasmids within the same isolate [3, 4].

There are two classification systems for β-lactamases that are currently in use. They include the Ambler molecular classification and the Bush, Jacoby, and Medeiros classification. In the Ambler classification, enzymes are further categorized into four classes A, B, C, and D enzymes based on conserved motifs and protein sequence. The Bush, Jacoby, and Medeiros classification groups β-lactamases according to their substrate and inhibitor profiles. The β-lactamase mediated resistance is either mediated by plasmid or expressed chromosomally. Nevertheless, the spread of β-lactamases has been reported to be linked frequently with plasmid mediated ESBLs, especially the CTX-M family [5, 6]. The CTX-M enzymes are under the class A group of enzymes in the Ambler classification of β-lactamases. ESBL-producing *Enterobacteriaceae* were first described in 1983 [7] in relation with hospital-acquired infections, with resistance arising from point mutations in enzymes that are plasmid mediated like TEM-1, TEM-2 and SHV-1 [1]. CTX-M enzymes are now predominant and have rapidly spread globally. The CTX-M enzymes have been divided into six groups (CTX-M-1, CTX-M-2, CTX-M-8, CTX-M-9, CTX-M-25, and KLUC, named after the first member of the group that was described). CTX-M-15 and CTX-M-14, a group 1 and 9 enzyme respectively are frequently identified globally in important microbes [8]. *Klebsiella pneumoniae* and *Escherichia coli* are the most common ESBL producing species. Transmission of ESBL-producing *Enterobacteriaceae* challenges healthcare facilities worldwide regarding the implementation of effective infection control measures to limit nosocomial spread. Recently, a global priority list of antibiotic resistant bacteria was published by the World Health Organization to increase and encourage research into new treatments for such pathogens. The list included extended-spectrum β-lactamase producing *Enterobacteriaceae* which calls for concern both in healthcare and community settings [6]. Extended-spectrum β-lactamase producing *Enterobacteriaceae* cause high morbidity and mortality rate and increased healthcare costs. An increasing prevalence of ESBL-Enterobacteriaceae correlates with a rise in the consumption of carbapenems [9, 10]; and this appears to have resulted in the emergence and spread of carbapenem resistance, especially in *Enterobacteriaceae* [11]. Carbapenemase-producing *Enterobacteriaceae* have been reported globally which includes a recent report of $bla_{OXA-48}$, $bla_{OXA-181}$, $bla_{NDM1}$ in *Enterobacteriaceae* from Edo state, Nigeria in which some of the isolates had ESBL genes specifically CTX-M-15 [12, 13, 14].

Many reports have described and characterized ESBLs in *Enterobacteriaceae* [2, 7], but only a few reports originate from African countries [15, 16, 17, 18, 19, 20]. The aim of this study was to determine the frequency of ESBL genes in multidrug resistant human clinical *Enterobacteriaceae* isolates from Edo state Nigeria and to characterize the resistance mechanisms using whole genome sequencing.

## Materials and methods

### Bacterial isolates

A total of 217 consecutive clinical isolates of *Enterobacteriaceae* were collected from March-May 2015 from three medical microbiology laboratories of hospitals all in Edo state Nigeria (Fig 1). Isolates obtained were from samples of both in-patients and out-patients of the three different hospitals. The three hospitals are tertiary healthcare institutions with University of Benin teaching hospital having the largest number of bed capacity of over 910. Identities of the isolates were confirmed by MALDI-TOF mass spectrometry (Bruker Daltonik GmbH, Bremen, Germany) analysis.

**Ethical considerations.** Research ethics approval was not required in the three hospitals where isolates were obtained from for the study. Only the pre-identified isolates used in the study were obtained from the microbiology laboratories. There was no contact with patients and the samples where the isolates were obtained from. Informed consent was not required by the institutions where the isolates were collected from as the data collected regarding the isolates which include patients' age, gender and sample source of isolates was obtained from clinical records and analyzed anonymously. Data obtained was for analysis purpose and this was carried out anonymously.

### Antimicrobial susceptibility testing and ESBL screening

The Kirby-Bauer susceptibility testing technique [21] was carried out and results were interpreted using European Committee on Antimicrobial Susceptibility Testing, (EUCAST) criteria

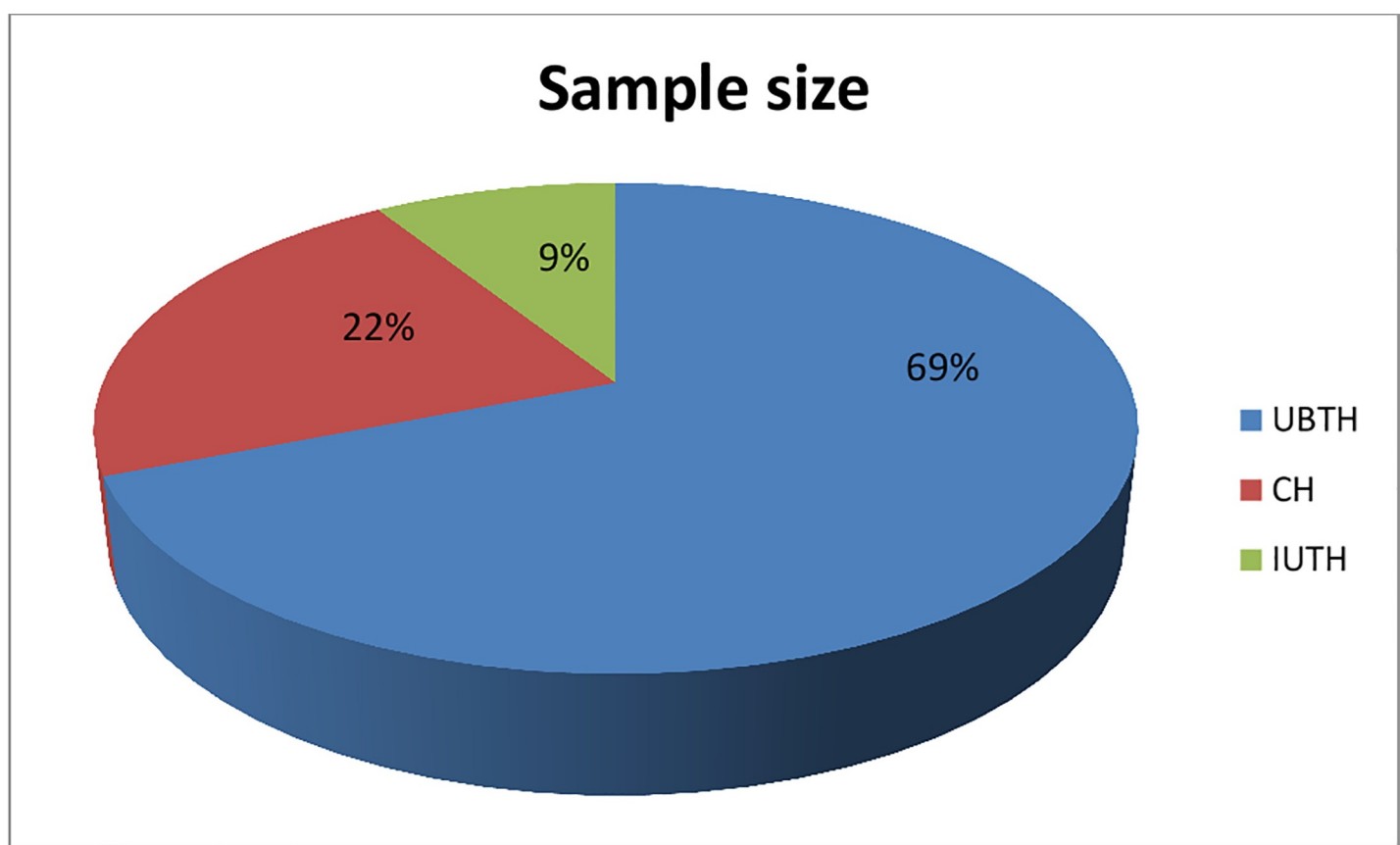

**Fig 1. Figure showing sample size of *Enterobacteriaceae* isolates provided by three Nigerian hospitals.**

[22]. The isolates were tested with 14 antibiotics: meropenem, ertapenem, ceftazidime, cefotaxime, amoxicillin/clavulanic acid, cefoxitin, cefepime, aztreonam, trimethoprim, ciprofloxacin, amikacin, piperacillin/tazobactam, chloramphenicol, levofloxacin (Oxoid, Basingstoke Hampshire, UK). The double disc synergy test was used to screen for the production of ESBLs [7]. ESBL production was screened phenotypically using cefotaxime (30 μg). The isolates were confirmed by combination disk tests with cefotaxime and ceftazidime (30 μg), with and without clavulanic acid (10 μg), as described by the CLSI guidelines [23]. A ≥ 5 mm increase in the zone diameter for cefotaxime or ceftazidime in combination with clavulanic acid versus the zone diameter of the corresponding antimicrobial agent alone defined an ESBL producer [23]. Isolates that were multidrug resistant and phenotypically positive for ESBL production were subjected to whole genome sequencing. Multidrug resistance was defined as non-susceptibility of an *Enterobacteriaceae* isolate to ≥1 agent of ≥3 antimicrobial categories [24]. Intrinsic resistances were not taken into account for categorization. The MICs of ampicillin, ciprofloxacin, trimethoprim were tested using E-test strips from bioMérieux (Marcy L'Etoile, France), those for amoxicillin-clavulanic acid, cefotaxime, cefepime, ceftazidime, aztreonam, amikacin using E-test strips from Oxoid (Basingstoke Hampshire, UK) for all the ESBL producing strains as previously described [25] and results were interpreted using EUCAST criteria [22].

## Whole genome sequencing

Whole genome sequencing (WGS) was performed for all 60 multidrug resistant isolates, which were also positive for the ESBL phenotypic test as described above. Extraction of genomic DNA (gDNA) was carried out using the MagAttract HMW DNA extraction kit (Qiagen, Hilden, Germany). Genomic DNA was quantified on a Qubit® 2.0 Fluorometer using the dsDNA BR Assay kit (Invitrogen by Thermo Fisher Scientific, Waltham, MA, USA) and diluted to 0.2ng/μl as recommended (Illumina sample preparation guide). The Illumina Nextera XT DNA library preparation kit (Illumina Inc, San Diego, CA, USA) was used for preparation of fragment libraries of the bacterial genomes. 1ng of gDNA was used for DNA fragment library preparation (Illumina sample preparation guide). Miseq reagent kit v3 containing the reagent cartridge and flow cell was used for a paired end sequencing using a read length of 2x300bp on an Illumina Miseq (Miseq v3.0, Illumina Inc). Pooled libraries were loaded on the reagent cartridge. Samples were sequenced for minimum targeted coverage of 100-fold using Illumina's recommended standard protocols.

For sequence analysis the raw reads (FASTQ files) were trimmed at their 5' and 3' ends until an average base quality of 30 was reached in a window of 20 bases, and assembly was performed with Velvet version 1.1.04 [26] using optimized k-mer size and coverage cutoff values based on the average length of contigs with >1,000 bp. Assembled genomes were uploaded to the ResFinder 2.1 web server (http://www.genomicepidemiology.org) to identify resistance genes using a threshold of 98% identity (ID) [27]. All ESBL isolates were further typed by multilocus sequence typing (MLST; MLST 1.8, Centre for Genomic Epidemiology (CGE), Lyngby DK). The identity of ESBL *Enterobacteriaceae* isolates were further confirmed by performing a tetra correlation search (TCS) between draft genomes obtained in this study and JSpeciesWS reference database based on their resulting Tetra-nucleotide signature correlation index [28]. PlasmidFinder 1.3 webtool (http://cge.cbs.dtu.dk/services/PlasmidFinder) was used to analyze and classify plasmids on the draft genomes of the ESBL *Enterobacteriaceae* isolates using a threshold of 95% ID [29]. Plasmid replicons were detected and plasmids were assigned to Inc groups.

## Nucleotide sequence accession numbers

This Whole Genome Shotgun project PRJNA600702 has been deposited at DDBJ/ENA/GenBank under the accession numbers JAAARR000000000-JAAATV000000000.

**Table 1. Antibiotic susceptibility test results showing % of resistant ESBL isolates.**

| Antibiotics | K. pneumoniae [n = 26] | E. coli [n = 17] | E. asburiae [n = 3] | E. cloacae [n = 5] | P.mirabilis [n = 1] | A.hermanii [n = 1] | C.werkmanii [n = 1] | E.hormaechei [n = 6] |
|---|---|---|---|---|---|---|---|---|
| Ceftazidime | 100%[n = 26] | 88%[n = 15] | 100%[n = 3] | 100%[n = 5] | - | 100%[n = 1] | 100%[n = 1] | 100%[n = 6] |
| Cefotaxime | 100%[n = 26] | 100% [n = 17] | 100%[n = 3] | 100%[n = 5] | 100%[n = 1] | 100%[n = 1] | 100%[n = 1] | 100%[n = 5] |
| Amoxicillin-clavulanic acid | 92%[n = 24] | 47%[n = 8] | 100%[n = 3] | - | - | - | 100%[n = 1] | - |
| Cefoxitin | 8%[n = 2] | 35%[n = 6] | 100%[n = 3] | 100%[n = 5] | - | - | 100%[n = 1] | 100%[n = 6] |
| Cefepime | 96%[n = 25] | 82%[n = 14] | 100%[n = 3] | 60%[n = 3] | - | 100%[n = 1] | 100%[n = 1] | 100%[n = 6] |
| Aztreonam | 100%[n = 26] | 94%[16] | 100%[n = 3] | 100%[n = 5] | - | 100%[n = 1] | 100%[n = 1] | 100%[n = 6] |
| Meropenem | 4%[n = 1] | - | - | - | - | - | - | - |
| Ciprofloxacin | 65%[n = 17] | 100% [n = 17] | 100%[n = 3] | 20%[n = 1] | 100%[n = 1] | - | 100%[n = 1] | 83%[n = 5] |
| Amikacin | 12%[n = 3] | - | - | 20%[n = 1] | - | - | - | - |
| Chloramphenicol | 39%[n = 10] | 59%[n = 10] | 100%[n = 3] | 100%[n = 5] | 100%[n = 1] | 100%[n = 1] | 100%[n = 1] | 67%[n = 4] |
| Piperacillin-tazobactam | 35%[n = 9] | 18[n = 3] | - | 40%[n = 2] | - | - | - | - |
| Ertapenem | 15%[n = 4] | - | - | - | - | - | - | 17%[n = 1] |
| Trimethoprim | 92%[n = 24] | 100% [n = 17] | 100%[n = 3] | 100%[n = 5] | 100%[n = 1] | 100%[n = 1] | 100%[n = 1] | 100%[n = 6] |
| Levofloxacin | 54%[n = 14] | 100% [n = 17] | 67%[n = 2] | 20%[n = 1] | 100%[n = 1] | - | 100%[n = 1] | 83%[n = 5] |

## Results and discussion

Out of 217 consecutive clinical *Enterobacteriaceae* isolates, 148 (68.2%) were multi-drug resistant and were further investigated by whole genome sequence (WGS) analysis (Table 1).

Of the multi-drug resistant isolates, sixty (40.5%) were positive for the ESBL phenotypic test and had ESBL genes detected. In total, the ESBL strains mostly were resistant to ceftazidime (88.9%), cefotaxime (93.7%), cefepime (82.5%) and aztreonam (90.5%). In addition, isolates showed resistance to amoxicillin-clavulanic acid (73%) and piperacillin-tazobactam (20.6%). MIC results confirmed data from the disc diffusion tests revealing high resistance rates among the ESBL producing isolates (S1 Table).

Whole genome sequencing revealed that 46 ESBL producing isolates harbored one single ESBL resistance gene while 14 isolates harbored double ESBL resistance genes. Ten *Klebsiella pneumoniae* isolates harbored the $bla_{CTX-M-15}$ gene; one *K. pneumoniae* isolate harbored the $bla_{SHV-28}$ gene, and one *K.pneumoniae* isolate had the $bla_{CTX-M-11}$ gene. Thirteen *Klebsiella pneumoniae* isolates harbored both $bla_{CTX-M-15}$ and $bla_{SHV-28}$ gene and one *Klebsiella pneumoniae* isolate harbored both $bla_{CTX-M-15}$ and $bla_{SHV-33}$ gene. The $bla_{CTX-M-15}$ gene was also detected in 15 *E. coli*, three *Enterobacter asburiae*, five *Enterobacter cloacae*, six *Enterobacter hormaechei*, one *Atlantibacter hermannii*, one *Citrobacter werkmanii* and one *Proteus mirabilis* respectively. One *E. coli* harbored the ESBL gene $bla_{CTX-M-65}$ and one *E. coli* isolate had the $bla_{CTX-M-14}$. S1 Table shows the characteristics of the ESBL producing isolates. Multilocus sequence typing (MLST) grouped the 26 ESBL producing *K. pneumoniae* isolates into 14 known sequence types (ST) with ST 15 being predominant (n = 6), ST 14 (n = 4), ST 340 and ST 1552 (n = 2) respectively. The 17 ESBL producing *E. coli* isolates were grouped into 8 known STs with ST 405 being predominant (n = 5), ST 410 (n = 2), ST 359 (n = 1), ST 5869 (n = 1), ST 156, ST 617, ST 648 and ST 131 (n = 2) respectively. MLST grouped the five *E. cloacae* isolates into the 3 known sequence types, ST 187, ST 156 and ST 836. Two ESBL producing

*E. cloacae* were assigned to new sequence types. The six *Enterobacter hormaechei* were grouped into three different sequence types, ST 78, ST114 and ST 459 (n = 2) respectively. Eleven plasmid incompatibility groups were identified among the ESBL *K. pneumoniae* isolates which included the FIB, FII, FIA, P, Q1, R, HI1B, $X_3$, ColKP3, ColpVC, Col (BS512) plasmid replicon types. The ESBL producing *E. coli* had 14 plasmid incompatibility groups which included the FIB, FII, FIA, FIB(AP001918), FIB(pB171), Y, I1, FII(pRSB107), $X_4$, Col156, Col(MG828), FII (pCoo), Q1, Col (BS512) plasmid replicon types. All ESBL producing *E. asburiae* isolates had the FII(Yp), FIB(K) and P plasmid replicon types. Seven plasmid incompatibility groups were identified among the ESBL producing *E. cloacae* and *E. hormachei* isolates which included FII (Yp), R, FIB(K), FII(pECLA), HI2, HI2A, and A/C2 plasmid replicon types. The ESBL *A. hermannii* isolate had the HI2, HI2A plasmid replicon types while the *C. werkmanii* isolate had the FIB (pHCM2) and R plasmid replicon types. The ESBL producing *P. mirabilis* isolate yielded a Q1 plasmid replicon type.

ESBL producing organisms are an increasingly important cause of multi-drug resistant infections [30]. ESBL genes are often located within mobile genetic elements such as plasmids, transposons and integrons which also can contain other resistance genes, thereby conferring resistance to antimicrobials that are extensively used in animals and humans (e.g. trimethoprim, aminoglycosides, fluoroquinolones, and sulfonamides). This could play an important role in co-selection [31].

The number of multidrug resistant ESBL producing *Enterobacteriaceae* isolates in our study was 60 (40.5%). The $bla_{CTX-M-15}$ gene was identified in 93.3% (n = 56/60) of the ESBL producing isolates. This is consistent with previous findings demonstrating the predominance of CTX-M enzymes and in particular CTX-M-15 [32, 33]. CTX-M-15 enzymes have been identified worldwide [34] including Nigeria in *Klebsiella* spp., *Escherichia coli*, *Proteus* spp., *Pseudomonas aeruginosa* and *Enterobacter* spp. [19, 35, 36]. There is limited data on the prevalence of ESBL-producing *Enterobacteriaceae* in Africa. A previous report showed a high prevalence of ESBL carriage with multiple clones in children presenting at a pediatric department in Guinea-Bissau [17]. Another report from South Africa revealed the dissemination of ESBL-producing *K.penumoniae* within and between wards of hospitals [15]. In Angola, a high occurrence of CTX-M-15 in diverse *Enterobacteriaceae sp* and non-clinical niches was reported which included *Enterobacter hormaechei* (n = 2) and *Citrobacter werkmanii* (n = 1) [20]. Recently, a high prevalence of ESBL-producing isolates was reported among *K. pneumoniae* strains detected in clinical specimens from a teaching hospital in Côte d'Ivoire with paediatric patients being most frequently affected [18]. Previous reports also from Nigeria revealed ESBL production mediated by $bla_{VEB}$ in *Providencia* spp. isolated from chicken faecal samples [37] and ESBL production mediated by $bla_{VEB}$ in human clinical *Providencia* spp. and *Citrobacter freundii* isolates [36, 38]. To our knowledge this study presents the first genomic characterization of ESBL production mediated by $bla_{CTX-M-15}$ in human clinical isolates of *Enterobacter hormaechei*, *Citrobacter werkmanii* and *Atlantibacter hermannii* from Nigeria. Our results show a wide dissemination of the $bla_{CTX-M-15}$ gene, which could also be attributed to widespread and uncritical use of beta-lactam antibiotics. The results from this study are similar to previous studies especially from African countries where the increasing prevalence of ESBL *Enterobacteriaceae* especially $bla_{CTX-M-15}$ has been reported [15, 16, 17, 18, 19, 20]. Reports from developing countries especially in Africa are alarming considering the diagnostic and treatment possibilities available. Multiple resistance determinants were found on the draft genome sequences of the multidrug resistant ESBL producing isolates. In this study, 87% (n = 52/60) of ESBL producers harbored PMQR determinants thereby supporting the strong association between ESBL production and quinolone resistance previously reported in *Enterobacteriaceae* [39]. The *aac(6')-lb-cr* gene has spread rapidly among *Enterobacteriaceae*, and

although conferring only low level resistance, it may create an environment facilitating the selection of highly resistant determinants especially in organisms harboring topoisomerase mutations. Resistance genes are also disseminated through horizontal gene transfer which is supported by the co-selection through various antimicrobials [39, 40]. The acquisition and accumulation of resistance determinants led to the emergence of multi-drug resistant ESBL producers further limiting therapeutic options.

ESBL genes of the TEM, SHV and CTX-M families can reside on large conjugative plasmids [7]. Additional resistance genes present on the same plasmid carrying $bla_{CTX-M-15}$ explain the multiresistant phenotype of CTX-M 15-producing bacteria [41]. Conjugative plasmids are regarded as one of the main factors in the successful spread of CTX-M type ESBLs among *Enterobacteriaceae* [42]. The presence of different plasmid replicon types in ESBL producing isolates reveal their importance in the dissemination of these resistance genes. The IncF family is the most commonly reported among the ESBL isolates. IncF plasmids are low-copy number plasmids, often carrying more than one replicon [43]. Detection of this plasmid type known to evolve quickly by replicon diversification and acquisition on antibiotic resistance traits [44] increases the potential role of *Enterobacteriaceae* as a reservoir for ESBL genes and other resistance determinants.

Application of whole genome sequencing offers a level of discrimination and information on relatedness and resistance mechanisms that surpass previous typing methods [45, 46]. ST 15, predominant among the ESBL producing *K. pneumoniae* isolates, corresponds to an internationally occurring clone and has been associated with different ESBL genes and MBL genes coding for NDM and VIM [47, 48]. Recently, Izdebski et al., [49] found that ST66, ST78, ST108 and ST114 strains spread as high-risk international clones of extended spectrum cephalosporin-resistant *E. cloacae*. The results from the MLST analysis of this study showed two *E. hormaechei* isolates belonged to the international clone ST114 recognized as a high-risk lineage responsible for the dissemination of carbapenemases and ESBLs in hospital setting [49]. For the *E. coli* isolates, MLST analysis revealed eight different known STs, including ST131, ST405 and ST410. ST131 is a dominant international clinical clone. ST405 has been associated internationally with the carriage of ESBLs and ST410 has been reported worldwide in extra intestinal pathogens associated with resistance to fluoroquinolones, third-generation cephalosporins and carbapenems [50, 51]. The distribution of the different sequence types and the plasmid replicon types of the ESBL isolates especially the predominant ones across the three hospitals suggests a probable clonal spread of the ESBL strains both intra and inter-hospital. A limitation of this study however, is to conclude on clonal bacterial transmission between ESBL *Enterobacteriaceae* isolates that included seven groups of different organisms. Another limitation of this study is the lack of adequate clinical data [e.g. differentiating samples of in-patients from out-patients] that might help to identify if the frequency of ESBL carriage was community based or nosocomial.

Previously, only a few reports from Nigeria provided draft genome sequence analyses, mainly concerning multidrug-resistant *E. coli* strains isolated from humans and chicken [52, 53]. Whilst three ESBL *E. coli* poultry isolates belonged to ST6359 [52], the poultry *E. coli* isolates reported by Sharma et al., were assigned to ST131 and ST162 respectively. The human *E. coli* isolates were assigned to six different STs which included ST131, ST617, ST542, ST398, ST4143 and ST398 [53].

## Conclusions

Many factors can contribute to high rates of antibiotic resistance in developing countries. These include: poor drug quality or inadequate posology, long-term treatments, misuse of

antibiotics by health professionals, unskilled practitioners, self-medication (antibiotics can be purchased without prescription), unhygienic conditions accounting for the spread of resistant bacteria and inadequate surveillance programs [54, 55]. In Nigeria the collapse of the primary healthcare system coupled with the unavailability of drugs in hospitals had the effect that people often resort to purchase drugs over the counter and in some cases from roadside sellers, which exposes them to the danger of acquiring and selecting ESBL-producing organisms [56]. Unlike the situation in developed countries, the financial resources to provide alternative agents such as carbapenems are lacking in developing countries and the option to tailor therapy based on antimicrobial resistance testing is unavailable, except for a few hospitals. To control the emergence and spread of ESBL-producing *Enterobactericeae*, it is essential for the public to practice good hygiene habits and comply with recommendations on the proper use of antibiotics. Proper infection-control practices and barriers are essential to prevent spreading and outbreaks of ESBL-producing bacteria. At an institutional level, practices that can minimize the spread of such organisms include clinical and bacteriological surveillance of patients admitted to intensive care units and antibiotic cycling, as well as policies of restriction, especially on the empirical use of broad-spectrum antimicrobial agents such as the third- and fourth-generation cephalosporins and quinolones [57].

Results from this study could serve a step towards developing targeted strategies to control the spread of ESBL producing bacteria in Benin city and Nigeria at large. Methods should be improved to efficiently detect and track bacterial clones and plasmids that constitute the major vehicles for the spread of ESBL-mediated resistance. Ideally, such methods of detection should be accessible to medium-level diagnostic microbiology laboratories, to assure the possibility of performing interventions in real time.

## Supporting information

**S1 Table. Characteristics of the ESBL isolates.**
(XLSX)

**S2 Table. Antibiotic susceptibility test result of the ESBL isolates.**
(DOCX)

**S3 Table. Source data of the ESBL isolates.**
(DOCX)

**S4 Table. Sequence and Plasmid replicon types of the ESBL isolates.**
(XLSX)

## Author Contributions

**Conceptualization:** Christiana Jesumirhewe.

**Investigation:** Christiana Jesumirhewe.

**Methodology:** Christiana Jesumirhewe.

**Supervision:** Burkhard Springer, Franz Allerberger, Werner Ruppitsch.

**Writing – original draft:** Christiana Jesumirhewe.

**Writing – review & editing:** Christiana Jesumirhewe, Burkhard Springer, Franz Allerberger, Werner Ruppitsch.

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
