## [Decision Letter · Decision Letter 0]

11 Nov 2019

PONE-D-19-26950

Molecular characterization of extended-spectrum β-lactamase genes in Enterobacteriaceae isolates from Nigeria

Dr Christiana Jesumirhewe, Department of Pharmaceutical microbiology, Igbinedion University Okada, Edo state, Nigeria

PLOS ONE

Dear Dr jesumirhewe,

Thank you for submitting your manuscript to PLOS ONE. After careful consideration, we feel that it has merit but does not fully meet PLOS ONE’s publication criteria as it currently stands. Therefore, we invite you to submit a revised version of the manuscript that addresses the points raised during the review process.

We would appreciate receiving your revised manuscript by Dec 26 2019 11:59PM. To enhance the reproducibility of your results, we recommend that if applicable you deposit your laboratory protocols in protocols.io, where a protocol can be assigned its own identifier (DOI) such that it can be cited independently in the future. For instructions see: http://journals.plos.org/plosone/s/submission-guidelines#loc-laboratory-protocols

We look forward to receiving your revised manuscript.

Kind regards,

Monica Cartelle Gestal, PhD

Academic Editor

PLOS ONE

Additional Editor Comments:

This manuscript is of great interest for us, please address the questions and suggestions offered by the reviewers.

2. Please amend either the title on the online submission form (via Edit Submission) or the title in the manuscript so that they are identical.

Reviewers' comments:

Reviewer's Responses to Questions

**Comments to the Author**

1. Is the manuscript technically sound, and do the data support the conclusions?

Reviewer #1: Yes

Reviewer #2: Partly

2. Has the statistical analysis been performed appropriately and rigorously? 

Reviewer #1: N/A

Reviewer #2: Yes

3. Have the authors made all data underlying the findings in their manuscript fully available?

Reviewer #1: No

Reviewer #2: Yes

4. Is the manuscript presented in an intelligible fashion and written in standard English?

Reviewer #1: Yes

Reviewer #2: No

5. Review Comments to the Author

Reviewer #1: Jesumirhewe and collaborators present an interesting manuscript with the general objective of determine the frequency of ESBL genes in multidrug resistant human clinical Enterobacteriaceae isolates from Edo state Nigeria and to characterize the resistance mechanisms. Thank you for being able to contribute to it before your presentation to the scientific community. Therefore, I suggest that the following questions be modified, deleted, or explained.

Line 1. I think authors should change in the title: Whole genome sequence of extended-spectrum β-lactamase genes in Enterobacteriaceae isolates from Nigeria.”

Improve the Abstract and Introduction. ESBL concern in both healthcare and community settings

Line 16, 64. “This study aimed at determining the frequency of ESBL genes in multidrug resistant human clinical Enterobacteriaceae isolates from Edo state Nigeria” using whole genome sequencing

Line 33. I suggest to describe Ambler classification of beta-lactamases and groups of CTX-M on the introduction.

Line 51. I would rather see table 1 summarized in a Figure instead of a table. I suppose it would be better demonstrated. The three hospitals are at a different level of care?

Why did you decide to include the strains of unidentified source; Teaching Hospital, Benin n=12, and Okada n=4

Line 74. “Flurometer” Fluorometer

Line 94. The authors reported that (41%) were positive for the ESBL phenotypic test and had ESBL genes detected. Please explain the results of Pret3577 and EC11, where no resistance pattern to B lactams was observed. In strains with ESBL and carbapenemase, the ESBL screening and confirmation performs well?

Line 115. Table 4. Please highlight in results the presence of carbapenemases NDM and Oxa-48 genes in strains Kp_852 ST15, Kp_852K ST15, Kp_872 ST340 (First cases described?)

I recommend to improve tables, columns of ST set in the beginning

Have all the sequences been deposited at GenBank?

Is it possible probable clonal spread of the ESBL strains Intra or inter-hospital

Limitations must be included

I suggest to include virulence factors and mobile genetic elements associated with ESBL and phylogeny based on core genome MLST.

I recommend to highlight the WGS findings (figures), highlight the emerging global high risk clones, and to present this manuscript in a better fashion.

Reviewer #2: The manuscript evaluates the frequency of ESBL genes in multidrug resistant human clinical Enterobacteriaceae isolates from Edo state Nigeria and characterize the resistance mechanisms. The research is correctly conducted, and the conclusions are supported by the data. However, results are not well exposed and are difficult to understand.

Materials and method

Line 66. The phenotypic method used, is not useful for microorganisms with Amp C , so some ESBL producers could not be included, and this could be considered as a limitation

Table 1. Use italics to Enterobacteriaceae. This table does not present summarized data and contains partial results. Include n values

Results: Include summary tables or figures

Table 2. Present information by isolated microorganism or at least the predominant ones. cite the % and include the n, also include the denominator of isolates tested for each antimicrobial listed

Line 93. Include n value

Table 3. summarize with MIC 50 and MIC 90 values. This information must be included as supplementary material.

Line 99 -114 summarize using tables or figures

Line 139. How many isolates were studied by MLST? Include n values

What was the resistance mechanism involved in meropenem resistance?

None S. marcescens isolate had ESBL genes?

6. PLOS authors have the option to publish the peer review history of their article (what does this mean?). If published, this will include your full peer review and any attached files.

Reviewer #1: Yes: José E Villacís

Reviewer #2: No

---

## [Author Response · Author response to Decision Letter 0]

7 Jan 2020

Additional Editor’s comments

This manuscript is of great interest for us, please address the questions and suggestions offered by the reviewers.

The manuscript has been re-written to meet PLOS ONE’s style requirements including those for file naming.

2. Please amend either the title on the online submission form (via Edit Submission) or the title in the manuscript so that they are identical.

The title on the online submission form and the manuscript are now identical.

All necessary data are made available already and relevant accession numbers are provided in the revised manuscript.

The supporting information files have been worked on and in-text citations updated to match accordingly

Review Comments to the Author

Reviewer #1: Jesumirhewe and collaborators present an interesting manuscript with the general objective of determine the frequency of ESBL genes in multidrug resistant human clinical Enterobacteriaceae isolates from Edo state Nigeria and to characterize the resistance mechanisms. Thank you for being able to contribute to it before your presentation to the scientific community. Therefore, I suggest that the following questions be modified, deleted, or explained.

Line 1. I think authors should change in the title: Whole genome sequence of extended-spectrum β-lactamase genes in Enterobacteriaceae isolates from Nigeria.

The title of the article has been changed to Whole genome sequencing of extended-spectrum β-lactamase genes in Enterobacteriaceae isolates from Nigeria

Improve the Abstract and Introduction. ESBL concern in both healthcare and community settings

The abstract and introduction has been improved on. Concern about ESBL in both healthcare and community settings have been highlighted

Line 16, 64. “This study aimed at determining the frequency of ESBL genes in multidrug resistant human clinical Enterobacteriaceae isolates from Edo state Nigeria” using whole genome sequencing

The sentence has been edited in the revised manuscript

Line 33. I suggest to describe Ambler classification of beta-lactamases and groups of CTX-M on the introduction.

Ambler classification of beta-lactamases have been described and groups of CTX-M mentioned on the introduction

Line 51. I would rather see table 1 summarized in a Figure instead of a table. I suppose it would be better demonstrated. The three hospitals are at a different level of care?

Table 1 is now presented as a figure. The 3 hospitals are tertiary healthcare institutions.

Why did you decide to include the strains of unidentified source; Teaching Hospital, Benin n=12, and Okada n=4

I included strains of unidentified source [Data not available} because some of these strains are important i.e had the esbl resistant gene and associated resistant determinants. Removing them from the number of isolates would not give a true picture of the frequency of resistant determinants present in the sample size 

Line 74. “Flurometer” Fluorometer

Fluorometer spelling corrected

Line 94. The authors reported that (41%) were positive for the ESBL phenotypic test and had ESBL genes detected. Please explain the results of Pret3577 and EC11, where no resistance pattern to B lactams was observed. In strains with ESBL and carbapenemase, the ESBL screening and confirmation performs well?

There was a mix-up in our earlier report. Isolates that were positive for the ESBL phenotypic test and had ESBL genes detected are sixty [40%]. Pret3577 was sensitive to the ESBL phenotypic test but had the ESBL gene detected by WGS. EC 11 was resistant to both ceftazidime and cefotaxime. The ESBL screening and confirmation performed well in strains with ESBL and carbapenemase

Line 115. Table 4. Please highlight in results the presence of carbapenemases NDM and Oxa-48 genes in strains Kp_852 ST15, Kp_852K ST15, Kp_872 ST340 (First cases described?)

I recommend to improve tables, columns of ST set in the beginning

Have all the sequences been deposited at GenBank?

Is it possible probable clonal spread of the ESBL strains Intra or inter-hospital

Limitations must be included

I suggest to include virulence factors and mobile genetic elements associated with ESBL and phylogeny based on core genome MLST.

I recommend to highlight the WGS findings (figures), highlight the emerging global high risk clones, and to present this manuscript in a better fashion.

The carbapenemases have been earlier reported along with a few other isolates that are not ESBL producers as indicated in the manuscript with tracked changes. The aim of this study was to determine the frequency of ESBL genes in multidrug resistant human clinical Enterobacteriaceae isolates from Edo state Nigeria. Tables have been improved on. Table S1 gives the characteristics of the ESBL isolates. Sequences have been deposited at the GenBank and accession numbers included on Table S1.Yes there is a possibility of probable clonal spread of the ESBL strains especially intra-hospital. The WGS findings have been re-presented. A limitation in this study is to conclude on clonal bacterial transmission between the ESBL Enterobacteriaceae isolates that included seven groups of different organisms. However, the presence of different plasmid replicon types in ESBL producing isolates reveals their importance in the dissemination of these resistance genes.

Reviewer #2: The manuscript evaluates the frequency of ESBL genes in multidrug resistant human clinical Enterobacteriaceae isolates from Edo state Nigeria and characterize the resistance mechanisms. The research is correctly conducted, and the conclusions are supported by the data. However, results are not well exposed and are difficult to understand.

The results have been represented and are now easy to understand

Materials and method

Line 66. The phenotypic method used, is not useful for microorganisms with Amp C , so some ESBL producers could not be included, and this could be considered as a limitation

The phenotypic method used was based on CLSI guidelines. All ESBL producers detected phenotypically were confirmed in the WGS.Only one isolate had ESBL detected by WGS but not detected phenotypically

Table 1. Use italics to Enterobacteriaceae. This table does not present summarized data and contains partial results. Include n values

Table 1 has been re-presented as a figure

Results: Include summary tables or figures

Results have been re-presented

Table 2. Present information by isolated microorganism or at least the predominant ones. cite the % and include the n, also include the denominator of isolates tested for each antimicrobial listed

Information has been re-presented showing the % of ESBL isolates resistant to the various antibiotics with n included and the denominator of isolates tested for each antimicrobial listed

Line 93. Include n value

n value has been included

Table 3. summarize with MIC 50 and MIC 90 values. This information must be included as supplementary material.

The MIC values are included as supporting information now.The MIC values are presented only for the 60 ESBL isolates that are multidrug resistant. MIC values were not determined for the entire 218 enterobacteriaceae isolates. The E-test has been reported to be considered a reliable method to determine antimicrobial susceptibility testing and it gives results which are at least as accurate as those obtained by the broth dilution method. As our ESBL isolates are not up to 100 and the Enterobacteriaceae isolates are a collection of 7 different group of isolates calculating MIC50 and MIC90 may not be applicable here

Line 99 -114 summarize using tables or figures

Line 99-114 has been summarized and tables represented. The genetic characteristics of the ESBL isolates are presented in the supporting information with other details.

Line 139. How many isolates were studied by MLST? Include n values

All the multidrug resistant isolates that are ESBL producers were studied by MLST, n=60, The sensitive isolate that genotypically had ESBL detected was also studied by MLST, n=1.Some of the isolates did not have available sequence types.

What was the resistance mechanism involved in meropenem resistance?

None S. marcescens isolate had ESBL genes?

No S. marcescens isolate had ESBL genes. The carbapenemases have been earlier reported along with a few other isolates that are not ESBL producers as indicated in the manuscript with tracked changes. The aim of this study was to determine the frequency of ESBL genes in multidrug resistant human clinical Enterobacteriaceae isolates from Edo state Nigeria.

---

## [Decision Letter · Decision Letter 1]

4 Feb 2020

PONE-D-19-26950R1

Whole genome sequencing of extended-spectrum β-lactamase genes in Enterobacteriaceae isolates from Nigeria 

PLOS ONE

Dear Dr jesumirhewe,

Thank you for submitting your manuscript to PLOS ONE. After careful consideration, we feel that it has merit but does not fully meet PLOS ONE’s publication criteria as it currently stands. Therefore, we invite you to submit a revised version of the manuscript that addresses the points raised during the review process.

We would appreciate receiving your revised manuscript by Mar 20 2020 11:59PM. To enhance the reproducibility of your results, we recommend that if applicable you deposit your laboratory protocols in protocols.io, where a protocol can be assigned its own identifier (DOI) such that it can be cited independently in the future. For instructions see: http://journals.plos.org/plosone/s/submission-guidelines#loc-laboratory-protocols

We look forward to receiving your revised manuscript.

Kind regards,

Monica Cartelle Gestal, PhD

Academic Editor

PLOS ONE

Reviewers' comments:

Reviewer's Responses to Questions

**Comments to the Author**

1. If the authors have adequately addressed your comments raised in a previous round of review and you feel that this manuscript is now acceptable for publication, you may indicate that here to bypass the “Comments to the Author” section, enter your conflict of interest statement in the “Confidential to Editor” section, and submit your "Accept" recommendation.

Reviewer #1: All comments have been addressed

Reviewer #2: All comments have been addressed

2. Is the manuscript technically sound, and do the data support the conclusions?

Reviewer #1: Partly

Reviewer #2: Yes

3. Has the statistical analysis been performed appropriately and rigorously? 

Reviewer #1: N/A

Reviewer #2: Yes

4. Have the authors made all data underlying the findings in their manuscript fully available?

Reviewer #1: Yes

Reviewer #2: Yes

5. Is the manuscript presented in an intelligible fashion and written in standard English?

Reviewer #1: Yes

Reviewer #2: Yes

6. Review Comments to the Author

Reviewer #1: Line 36. Enterobacteriaceae

Line 38. Specie of Providencia spp

Line 88. Repeated “Fig 1. Figure showing sample size of Enterobacteriaceae isolates provided by three Nigerian hospitals”

Line 90. Incomplete sentence

Line 94. patients’

According to the answer The ESBL screening and confirmation performed well in strains with ESBL and carbapenemase. Please consider that strains with ampC and carbapenemases can affect the performance of ESBL screening test

Line 87. Please include that the 3 hospitals are tertiary healthcare institutions, number of beds

Line 116- 128 Consider to rewrite the paragraph. There are words repeated “manufacturer’s instructions”

Line 195. Specie of Providencia spp

Line 233. The study concludes a probable clonal spread of the ESBL strains especially intra-hospital. Does WGS gave you some evidence? If not please consider the limitations.

The findings in the study can support a better conclusion in the abstract and the manuscript

Reviewer #2: The manuscript have been improved in quality, but there are some comments

Line 77. Please use additional bibliography to support this statement

Line 79. Please check the bibliography number 21

Line 87. Include that are tertiary hospitals. As a comment, Figure 1 does not improve the manuscript

I have some questions about isolates:

All isolates were obtained from hospitalized patients? Please specify if not

Line 134: specify which isolates were further studied with MLST. The results are just for K.pneumoniae, E.coli and E. cloacae?

Results

Piperacillin-tazobactam for K. pneumonia n = 9

results about imcompatibility groups and ST shoould be presented in a resume table. The current version is hard to read

Discussion

The manuscript does not highlight well their results. Include some data about ESBL in other countries from Africa and compare them.

7. PLOS authors have the option to publish the peer review history of their article (what does this mean?). If published, this will include your full peer review and any attached files.

Reviewer #1: Yes: José E Villacís

Reviewer #2: No

---

## [Author Response · Author response to Decision Letter 1]

25 Feb 2020

Reviewer No1

Line 36. Enterobacteriaceae

Correction has been effected on the revised manuscript

Line 38. Specie of Providencia spp

The revised manuscript has been updated. We decided to strike out Providencia spp as part of the reported ESBL isolates as it was sensitive to the ESBL phenotypic screening but we detected CTX-M-15 by sequencing. However during our submission of the sequences to NCBI also the sequence had issues that were challenging to resolve so we decided not to include it in the study anymore. The genome submission to NCBI gave a better insight with regards to the isolates.

Line 88. Repeated “Fig 1. Figure showing sample size of Enterobacteriaceae isolates provided by three Nigerian hospitals”

Correction has been effected on the revised manuscript.

Line 90. Incomplete sentence

Sentence has been updated in the revised manuscript.

Line 94. patients

Correction has been effected on the revised manuscript

According to the answer The ESBL screening and confirmation performed well in strains with ESBL and carbapenemase. Please consider that strains with ampC and carbapenemases can affect the performance of ESBL screening test

Noted. The 60 ESBL isolates presented including the ones with carbapenemases were positive to both the phenotypic screening for ESBL and the sequencing. Alongside other resistance genes only carbapenemases were detected in the ESBL isolates. No ampC was detected in any of the ESBL isolates.

Line 87. Please include that the 3 hospitals are tertiary healthcare institutions, number of beds

Revised manuscript has been updated

Line 116- 128 Consider to rewrite the paragraph. There are words repeated “manufacturer’s instructions”

Paragraph has been revised.

Line 195. Specie of Providencia spp

Manuscript has been revised. Providencia isolate is no longer included in the study. Reasons have been stated earlier in the rebuttal letter.

Line 233. The study concludes a probable clonal spread of the ESBL strains especially intra-hospital. Does WGS gave you some evidence? If not please consider the limitations.

Using the data obtained from the MLST assignment and plasmid incompatibility grouping, the distribution of the sequence types and the plasmid replicon types especially the predominant ones of the ESBL isolates across the three hospitals suggests a probable clonal spread of the ESBL strains both intra and inter-hospital. A limitation of this study however is to conclude on clonal bacterial transmission between ESBL Enterobacteriaceae isolates that included seven groups of different organisms.

The findings in the study can support a better conclusion in the abstract and the manuscript

Conclusion in the abstract and manuscript have been revised

Reviewer No 2

Line 77. Please use additional bibliography to support this statement

Additional bibliography has been added to the revised manuscript.

Line 79. Please check the bibliography number 21

Bibliography no 21 has been revised

Line 87. Include that are tertiary hospitals. As a comment, Figure 1 does not improve the manuscript

The manuscript has been revised indicating the hospitals are tertiary healthcare institutions. The figure was included in response to another reviewer’s comment from the previous revision.

All isolates were obtained from hospitalized patients? Please specify if not

Line 134: specify which isolates were further studied with MLST. The results are just for K.pneumoniae, E.coli and E. cloacae?

Isolates were obtained from samples of both in-patients and out-patients of the three different hospitals. All ESBL isolates were further studied with MLST. Sequence type numbers were not available for other isolates not presented. Some isolates had unknown sequence types which are submitted for curation of sequence type number.

Results

Piperacillin-tazobactam for K. pneumonia n = 9

Correction has been updated in the revised manuscript

results about imcompatibility groups and ST shoould be presented in a resume table. The current version is hard to read

Table has been revised. Result about incompatibility groups and ST has been represented in a separate table.

Discussion

The manuscript does not highlight well their results. Include some data about ESBL in other countries from Africa and compare them.

 Discussion has been revised highlighting the results properly. Data about ESBL in other countries from Africa have been included and compared with the study

---

## [Decision Letter · Decision Letter 2]

18 Mar 2020

Whole genome sequencing of extended-spectrum β-lactamase genes in Enterobacteriaceae isolates from Nigeria

PONE-D-19-26950R2

Dear Dr. jesumirhewe,

We are pleased to inform you that your manuscript has been judged scientifically suitable for publication and will be formally accepted for publication once it complies with all outstanding technical requirements.

With kind regards,

Monica Cartelle Gestal, PhD

Academic Editor

PLOS ONE

Additional Editor Comments (optional):

Reviewers' comments:

Reviewer's Responses to Questions

**Comments to the Author**

1. If the authors have adequately addressed your comments raised in a previous round of review and you feel that this manuscript is now acceptable for publication, you may indicate that here to bypass the “Comments to the Author” section, enter your conflict of interest statement in the “Confidential to Editor” section, and submit your "Accept" recommendation.

Reviewer #1: All comments have been addressed

2. Is the manuscript technically sound, and do the data support the conclusions?

Reviewer #1: Yes

3. Has the statistical analysis been performed appropriately and rigorously? 

Reviewer #1: N/A

4. Have the authors made all data underlying the findings in their manuscript fully available?

Reviewer #1: Yes

5. Is the manuscript presented in an intelligible fashion and written in standard English?

Reviewer #1: Yes

6. Review Comments to the Author

Reviewer #1: Line 36-37. The report of blaCTX-M-15 in clinical isolates of Providencia spp., Citrobacter freundii and Atlantibacter hermannii is important, however the results of WGS, plasmid profiles an ST related to the dissemintation of CTX-M are more relevant.

Line 79. Please include what carbapenemases were recently reported. KPC, NDM, OXA-48 ???

7. PLOS authors have the option to publish the peer review history of their article (what does this mean?). If published, this will include your full peer review and any attached files.

Reviewer #1: Yes: José E Villacís

---

## [Editor Report · Acceptance letter]

31 Mar 2020

PONE-D-19-26950R2 

Whole genome sequencing of extended-spectrum β-lactamase genes in *Enterobacteriaceae* isolates from Nigeria 

Dear Dr. Jesumirhewe:

I am pleased to inform you that your manuscript has been deemed suitable for publication in PLOS ONE. Congratulations! Your manuscript is now with our production department. 

With kind regards,

on behalf of

Dr. Monica Cartelle Gestal 

Academic Editor

PLOS ONE